# Oxytocin Modifies the Excitability and the Action Potential Shape of the Hippocampal CA1 GABAergic Interneurons

**DOI:** 10.3390/ijms25052613

**Published:** 2024-02-23

**Authors:** Antonio Nicolas Castagno, Paolo Spaiardi, Arianna Trucco, Claudia Maniezzi, Francesca Raffin, Maria Mancini, Alessandro Nicois, Jessica Cazzola, Matilda Pedrinazzi, Paola Del Papa, Antonio Pisani, Francesca Talpo, Gerardo Rosario Biella

**Affiliations:** 1Department of Biology and Biotechnology “Lazzaro Spallanzani”, University of Pavia, 27100 Pavia, Italy; antonionicolas.castagno01@universitadipavia.it (A.N.C.); paolo.spaiardi@unipv.it (P.S.); arianna.trucco01@universitadipavia.it (A.T.); francesca.raffin01@universitadipavia.it (F.R.); alessandro.nicois@gmail.com (A.N.); jessica.cazzola02@universitadipavia.it (J.C.); matilda.pedrinazzi01@universitadipavia.it (M.P.); paola.delpapa01@universitadipavia.it (P.D.P.); 2Department of Brain and Behavioral Sciences, University of Pavia, 27100 Pavia, Italy; maria.mancini@unipv.it (M.M.); antonio.pisani@unipv.it (A.P.); 3IRCCS Mondino Foundation, 27100 Pavia, Italy; 4INFN—Pavia Section, 27100 Pavia, Italy; 5Department of Biotechnology and Biosciences, University of Milano-Bicocca, 20126 Milano, Italy; claudia.maniezzi@unimib.it; 6Institute of Biomolecular Chemistry, Consiglio Nazionale delle Ricerche (CNR), 80078 Pozzuoli, Italy; 7Department of Biomolecular Sciences, University of Urbino Carlo Bo, 61029 Urbino, Italy

**Keywords:** oxytocin, GABAergic interneurons, CA1, hippocampus, electrophysiology, patch-clamp, phase plot analysis, linear mixed-effects models

## Abstract

Oxytocin (OT) is a neuropeptide that modulates social-related behavior and cognition in the central nervous system of mammals. In the CA1 area of the hippocampus, the indirect effects of the OT on the pyramidal neurons and their role in information processing have been elucidated. However, limited data are available concerning the direct modulation exerted by OT on the CA1 interneurons (INs) expressing the oxytocin receptor (OTR). Here, we demonstrated that TGOT (Thr4,Gly7-oxytocin), a selective OTR agonist, affects not only the membrane potential and the firing frequency but also the neuronal excitability and the shape of the action potentials (APs) of these INs in mice. Furthermore, we constructed linear mixed-effects models (LMMs) to unravel the dependencies between the AP parameters and the firing frequency, also considering how TGOT can interact with them to strengthen or weaken these influences. Our analyses indicate that OT regulates the functionality of the CA1 GABAergic INs through different and independent mechanisms. Specifically, the increase in neuronal firing rate can be attributed to the depolarizing effect on the membrane potential and the related enhancement in cellular excitability by the peptide. In contrast, the significant changes in the AP shape are directly linked to oxytocinergic modulation. Importantly, these alterations in AP shape are not associated with the TGOT-induced increase in neuronal firing rate, being themselves critical for signal processing.

## 1. Introduction

Oxytocin (OT) is a small neuropeptide predominantly synthesized in the paraventricular nucleus and in the supraoptic nucleus of the hypothalamus [1]. The oxytocinergic neurons within these nuclei project to the posterior pituitary gland (neurohypophysis) for the secretion of OT into the bloodstream, so that it can carry out its peripheral functions, such as in parturition and lactation [2]. However, oxytocinergic projections also reach various brain areas, such as thalamus, cortex, amygdala, striatum, and hippocampus [3,4,5]. Within the central nervous system, OT serves as a potent neuromodulator, influencing a wide spectrum of social and emotional behaviors, including attachment, social recognition, social memory, aggression, and fear conditioning [6,7,8]. Consistently, deficits in the oxytocinergic system have been implicated in several neuropsychiatric disorders, including autism, schizophrenia, depression, bipolar disorder, and borderline personality disorder [9,10].

The hippocampus is a key structure for most of these emotional, behavioral, and cognitive functions [11,12,13]; therefore, many studies have explored the functional modulation exerted by the OT in this brain area. They have found that the precise action of the OT in the hippocampus is contingent upon a finely tuned, either direct or indirect, modulation of specific neurons within distinct hippocampal subregions [14]. Specifically, OT exerts its direct effects by binding the oxytocin receptor (OTR) that is selectively expressed by restricted classes of neurons, variable across the hippocampal subregions [14,15,16,17,18].

In CA1, OTR expression has been mainly reported in fast-spiking GABAergic interneurons (FS-INs) of the *stratum pyramidale* [19,20,21]. In particular, OT has been demonstrated to increase the firing rate of these neurons, with a consequent enhancement in the phasic and tonic inhibition of the CA1 pyramidal neurons and a reduction in their excitability [19,20]. However, these studies have predominantly focused on the indirect effects of OT on the projection pyramidal neurons, highlighting its impact on improving the signal-to-noise ratio and thereby enhancing the fidelity and temporal precision of information transfer through this area [19,20].

It should be highlighted that, to date, the direct oxytocinergic modulation of the CA1 interneurons (INs) expressing the OTR has not been studied in depth. It has not been clarified, for example, whether the increase in firing frequency is accompanied by changes in the excitability of these neurons (intended as the propensity of the neurons to discharge in response to the injection of a depolarizing current). Additionally, the impact of OT in shaping the action potentials (APs) remains unexplored in CA1 neurons. Notably, studies conducted in the CA2 region revealed modulations of the AP shape by OT [22], suggesting an additional way through which OT can alter cell physiology and signal processing. Therefore, in this study, we aimed at exploring the multiple modulatory effects exerted by TGOT (Thr4,Gly7-oxytocin), a selective oxytocin receptor agonist, on the CA1 OTR-expressing GABAergic INs. OTR is a G-protein-coupled receptor that can activate several intracellular signaling pathways that act on diverse effectors [23], which may result in multiple independent or interconnected effects. We then tried to identify the inter-relationships and potential causal links between the multiple modulations exerted by the peptide on the same cell, focusing on the electrophysiological outcome. For this purpose, the most widely used statistical methods in basic neuroscience (i.e., Student’s *t*-test, Pearson’s correlation analysis, …), which compare clustered data, cannot appropriately take into consideration the explicit and implicit dependencies between variables [24]. To account for data dependencies and adequately infer the presence/absence of causal dependencies between the data, we used linear mixed-effects models (LMMs). LMMs are a new statistical tool to model the linear relationship between a response variable and one or more explanatory variables considering both fixed and random effects. This methodology is gradually gaining importance in the neuroscience field, being particularly effective in ensuring robust analyses, reproducibility, and richer conclusions [24,25,26].

By elucidating the intricate mechanisms underlying the influence of OT on the CA1 OTR-expressing GABAergic INs, our aim was to provide a deeper understanding of the neuromodulatory role of this important and eclectic peptide within this critical area of the brain.

## 2. Results

### 2.1. OTR Activation Not Only Depolarizes and Increases the Firing Frequency of Specific GABAergic Interneurons in CA1 but Also Enhances Their Excitability

In a previous study, we demonstrated that the selective OTR agonist TGOT was able to depolarize and increase the firing frequency of specific CA1 OTR-expressing GABAergic INs [19]. Similar results were obtained by Owen et al. [20]. Here, we aimed at identifying the multiple modulations exerted by 1 µM TGOT on the CA1 OTR-expressing GABAergic INs. To selectively record from the GABAergic INs of the *stratum pyramidale*, in our experiments, we used a murine model featuring GABAergic INs labeled with GFP (GAD67-GFP^+^ (Δneo) mice). The location and the GABAergic identity of the recorded cells were confirmed a posteriori through biocytin labeling and imaging of the recorded cells, together with the intrinsic fluorescence of GAD67-GFP^+^ neurons (Figure 1A). To select the cells directly responding to the TGOT administration, we relied on the experimental procedures reported in the literature [19,20]. Specifically, we recorded 24 TGOT-responding CA1 GABAergic INs from 18 mice, the majority of which, based on their firing mode, were fast-spiking interneurons (FS-INs, *N* = 14/24). They showed a membrane capacitance (C_m_) of 64.1 ± 3.9 pF, an input resistance (R_in_) of 210 ± 19 MΩ, and a membrane resting potential (V_r_) of −71.3 ± 1.2 mV. By analyzing the effect of OTR activation on their membrane potential at the firing threshold, we found that the bath perfusion of 1 µM TGOT determined a depolarization of +2.22 ± 0.32 mV (*p* < 0.001) (Figure 1B,C) and a significant increase in the firing rate by a factor of 5.12 ± 1.01 (*p* < 0.001) (Figure 1B,D), as expected.

Then, we investigated whether TGOT could elicit changes in the excitability of the OTR-expressing INs. To this end, we recorded the voltage response to depolarizing injected current steps of increasing amplitude, starting from an imposed membrane potential of −70 mV both under control conditions and during TGOT perfusion. At the corresponding amplitude of injected current, the firing frequency in TGOT was always higher than that obtained under control conditions (Figure 2A). We quantified these differences by plotting the measured firing rate for each value of injected current versus the injected current itself (F–I plots). In seven out of eight analyzed cells, the F–I relationship obtained during TGOT perfusion was shifted to the left compared with that of the control, as shown in Figure 2B for a representative IN. Accordingly, we found that TGOT caused a significant decrease in the offset of the OTR-expressing INs (142 ± 24 pA in CTRL vs. 117 ± 16 pA in TGOT; *N* = 8 cells from five mice; *p* < 0.05), meaning a decrease in the minimum intensity of injected current required to obtain a firing response (Figure 2C). Note that the term “offset” was preferred to “rheobase” because the recordings in both TGOT and control conditions started from a membrane potential of −70 mV and not from the membrane resting potential of the cell, possibly affected by the depolarizing effect of TGOT. No difference was found in gain (i.e., the slope of the F–I relationship) (Figure 2D; 0.26 ± 0.05 Hz/pA in CTRL vs. 0.31 ± 0.05 Hz/pA in TGOT; *N* = 8 cells from five mice; *p* > 0.05).

Then, these data confirmed the depolarizing effect of the OT on the OTR-expressing CA1 GABAergic INs, which in turn caused an increase in their firing frequency. Furthermore, we found that OTR activation also resulted in a generalized enhancement in the cellular excitability, resulting in an increase in the firing rate in response to the same injected current.

### 2.2. OTR Activation Alters the Shape of the Action Potentials of the OTR-Expressing CA1 GABAergic Interneurons

Not only the frequency of the APs but also their shape could be modulated by OTR activation [22]. To test possible differences generated by TGOT bath-perfusion in parameters related to the shape of the APs in the OTR-expressing CA1 GABAergic INs, we selected 10 consecutive APs in CTRL and 10 consecutive APs in TGOT for each cell (*N* = 17 INs from 12 mice) to be examined via phase plot analysis [27,28]. The APs were selected based on the long-recording protocol described earlier, as shown in Figure 1B for a representative cell. The superposition of the sample APs in the control and TGOT showed clear changes in the shape at the level of the AP threshold, amplitude, afterhyperpolarization, and kinetics (Figure 3A). Phase–plane plots (dV/dt versus V) of the same APs also showed a marked difference (Figure 3B).

For the quantitative analysis, the following parameters were measured for each AP: the threshold (V_thr_), the amplitude (Amp), the peak value of the afterhyperpolarization (V_AHP_), the duration (Dur), the depolarization slope (i.e., the maximum depolarization kinetics; dV/dt_max_), and the repolarization slope (i.e., the maximum repolarization kinetics; dV/dt_min_). Before proceeding to the statistical analysis, the values relating to the 10 APs in CTRL and the 10 APs in TGOT were separately averaged for each cell to minimize the possible stochastic differences in the shape of the APs in each group. The V_thr_ showed a significant shift from −48.3 ± 0.9 mV in the CTRL toward more depolarized values (−47.1 ± 0.9 mV; *p* < 0.001) in response to TGOT (Figure 3C). This effect on the V_thr_ was accompanied by a reduction in the maximum voltage reached at the peak of the AP (V_peak_ = 9.11 ± 2.04 mV in the CTRL vs. 5.59 ± 2.06 mV in TGOT; *p* < 0.001), resulting in an overall significant decrease in the AP Amp (57.4 ± 2.3 mV in the CTRL, reduced to 52.7 ± 2.3 mV in TGOT; *p* < 0.001) (Figure 3C). V_AHP_ was also modulated toward less-hyperpolarized values in TGOT (−66.5 ± 1.0 mV in the CTRL vs. −64.6 ± 1.2 mV in TGOT; *p* < 0.001) (Figure 3C). Additionally, noteworthy changes occurred in the AP kinetics. The values of the dV/dt_max_ and the dV/dt_min_ decreased in absolute terms, with dV/dt_max_ shifting from 145 ± 13 mV/ms to 124 ± 12 mV/ms (CTRL vs. TGOT; *p* < 0.001) and dV/dt_min_ shifting from −70.2 ± 9.2 mV/ms to −60.8 ± 8.2 mV/ms (CTRL vs. TGOT; *p* < 0.001) (Figure 3D). Accordingly, overall, the AP Dur increased in the presence of TGOT (1.17 ± 0.14 ms in the CTRL vs. 1.24 ± 0.14 ms in TGOT; *p* < 0.001) (Figure 3E). The effects of TGOT became apparent approximately 200 s after the onset of drug perfusion. A comparison of the APs recorded in control conditions, spaced 200 s apart, revealed a slight reduction in V_peak_ of 1.54 ± 0.53 mV (*N* = 19, *p* < 0.05). In contrast, a more pronounced reduction in V_peak_ of 3.52 ± 0.64 mV was observed during TGOT perfusion (*N* = 19, *p* < 0.001). Thus, the reduction attributed to run-down was less prominent than that observed during TGOT conditions. This observation confirmed that the changes in AP shape were a result of an actual modulatory action of TGOT on the cell rather than being merely ascribable to run-down phenomena induced by the sustained firing over time.

The shape of the APs could influence the firing rate of the neurons, and this is discussed in more detail in the next paragraph. To couple AP shape with firing frequency, interspike intervals (ISIs) between consecutive APs were calculated (*N* = 10 ISIs for each condition per cell; *N* = 17 cells from 12 mice). The ISI represents the inverse of the instantaneous firing frequency and was significantly reduced in TGOT compared to control conditions (2.33 ± 0.49 s in CTRL vs. 0.49 ± 0.10 s in TGOT; *p* < 0.01) (Figure 3F), which is in line with the observed increase in the mean firing rate during drug perfusion (Figure 1D).

The shape of the APs is caused by the interplay of various ion currents and their respective gating properties. While direct recordings of these currents in voltage-clamp mode were not performed, we estimated their contribution from the APs recorded in current-clamp mode. By estimating the net ionic currents expressed during APs using the equation I_ionic_ = −C_m_ × (dV/dt) [28,29] (see Section 4), we were able to evaluate the changes induced by OTR activation on these currents (Figure 3G) (*N* = 17 cells from 12 mice). Specifically, the amplitude of the inward current during the AP was reduced in the presence of TGOT (−8894 ± 932 pA in the CTRL vs. −7533 ± 789 pA in TGOT; *p* < 0.001), as was the amplitude of the outward current (4280 ± 593 pA in the CTRL vs. 3669 ± 517 pA in TGOT; *p* < 0.001) (Figure 3H), consistent with the observed reductions in the depolarization and repolarization kinetics of the AP. Additionally, the area underlying the current, calculated as an integral and serving as an indicator of the net charge displaced during the event, was significantly decreased (inward area: −3518 ± 213 nC in the CTRL vs. −3199 ± 164 nC in TGOT; *p* < 0.01; outward area: 4342 ± 240 nC in the CTRL vs. 4016 ± 203 nC in TGOT; *p* < 0.001) (Figure 3I). These data clearly indicate that TGOT has a substantial influence in shaping the APs of the CA1 GABAergic INs modulated by it.

Many of the parameters that define the shape of the APs are intrinsically related to each other. We used Pearson correlation analysis to clarify the specific direct or inverse correlations between them. Being in the presence of repeated observations (i.e., 10 APs in CTRL and 10 APs in TGOT) for the same subject (i.e., cell), we computed the correlations at the subject level (i.e., we used the averages for each subject) to eliminate the subject effect in repeated measures [30,31]. Starting from the data obtained in control conditions (*N* = 17 cells from 12 mice), we plotted pairwise scatterplots of V_thr_, Amp, V_AHP_, dV/dt_max_, dV/dt_min_, and Dur (Figure 4A). Then, we calculated the Pearson correlation coefficients for each of them (Figure 4A inset). The same analyses were repeated for data derived in the presence of TGOT (Figure 4B). We found that the correlation matrices obtained in control and TGOT conditions were quite similar, although correlations were generally stronger under TGOT conditions. We managed to adequately interpolate virtually all the pairwise scatterplots with regression lines and found highly significant direct or inverse correlations in most of the parameters describing the AP shape (Figure 4), as expected. While describing the presence of correlations between the parameters, this analysis is inadequate for identifying causal dependencies between them and could not highlight the effect of TGOT in inducing variations in the associated variables. So, for a more coherent analysis and interpretation of the data, we relied on LMMs.

### 2.3. The Linear Mixed-Effects Models Corroborate the Results and Highlight the Presence/Absence of Interactions and Causal Dependencies among the Effects Induced by TGOT

LMMs are a powerful statistical tool because they consider dependencies between data, allowing for robust and consistent results. Furthermore, based on them, interesting inferences can be made about how multiple elements interact with each other to cause an effect [24].

One of the main problems in statistical testing is the determination of the correct sample size for analyses. In our experiments regarding the shape of the Aps, we analyzed 10 APs in the CTRL and 10 APs in the TGOT group for each neuron, for a total of 170 APs in the CTRL and 170 APs in TGOT out of 17 INs from 12 mice. These data can be considered “pseudoreplications” [32] since multiple APs were derived from the same cell in each condition. To avoid artificial inflation of the sample size, resulting in false positives [32], we averaged the data before comparing them with the *t*-test (Figure 3). However, this approach might not be the best option for the analysis of these data. Instead, utilizing LMMs, which take data dependencies into account, could offer a more suitable approach [24]. To test whether this was the case, we calculated the intraclass correlation coefficient (ICC) for our data. The ICC is a metric frequently employed to evaluate the degree of correlation between measurements within the same group. It can range from zero to one: the closer it is to one, the more the data are correlated; the closer it is to zero, the more the data are uncorrelated [24]. For our data, we found ICC values close to one for almost all the considered parameters (Table 1), suggesting a large intraclass homogeneity (i.e., homogeneity between data derived from the same cell) and interclass heterogeneity (i.e., heterogeneity between data derived from different cells). When this situation occurs, the data cannot be treated as independent. On the other hand, the use of the *t*-test on aggregate data runs the risk of providing incomplete and not entirely reliable information. LMMs overcome these problems. We then constructed LMMs for each variable of interest, using the CTRL/TGOT condition as a fixed effect. The *p*-values assessed through the LMMs using the *anova.lme* function (see Section 4) allowed us to corroborate the results of the previously conducted *t*-tests, confirming the significant effect of the TGOT on all the parameters under examination (Table 1).

OT is a peptide that can exert multiple modulatory effects through the activation of heterogeneous signaling mechanisms with different final targets and/or by initiating signaling cascades that lead to interconnected and causally related effects [14,23]. An aspect of interest in the evaluation of the modulatory effect of OT on neurons lies in the identification of the relationships that exist between the various parameters altered by the activation of the OTR. LMMs have proven to be instrumental in facilitating this type of evaluation, allowing the assessment of the presence or absence of causal links among the parameters modified by the bath application of TGOT. To this end, we constructed LMMs with TGOT and a selected AP parameter as fixed effects and another different AP parameter or the ISI as the dependent variable.

We started from the study of the influence of TGOT and dV/dt_max_ on the AP Amp, since it is plausible that a reduced influx of sodium ions (as estimated in Figure 3G–I) could lead to a decrease in the depolarization slope of the AP, which, in turn, could be reflected in the AP Amp. Moreover, dV/dt_max_ and Amp were found to be directly correlated through Pearson correlation analysis (Figure 4). The LMM indicated that Amp was directly influenced by dV/dt_max_ (as dV/dt_max_ decreases, meaning the depolarization kinetics slows down, Amp also decreases; *p* < 0.001), and that TGOT and dV/dt_max_ interacted constructively in determining an amplified effect on Amp (i.e., a more pronounced reduction of Amp; *p* < 0.001) (Figure 5A and Table 2). For similar reasons involving potassium ion efflux, we constructed a LMM to study the influence of TGOT and dV/dt_min_ on V_AHP_. We found that V_AHP_ was directly influenced by dV/dt_min_ (as dV/dt_min_ increases, meaning the repolarization kinetics slow down, V_AHP_ increases in turn, meaning it becomes less negative; *p* < 0.01), and that TGOT and dV/dt_min_ interacted constructively in determining an amplified effect on V_AHP_ (i.e., a more pronounced increase of V_AHP_; *p* < 0.001) (Figure 5B and Table 2). We then tested whether slowdowns in depolarization (dV/dt_max_) and repolarization (dV/dt_min_) kinetics would induce reductions in AP Dur, and how TGOT would interact. We found that dV/dt_max_ and dV/dt_min_ influenced Dur inversely (as the dV/dt_max_ decreases, i.e., the depolarization kinetics slows down, Dur increases; *p* < 0.001), and directly (as the dV/dt_min_ increases, i.e., the repolarization kinetics slows down, Dur increases in turn; *p* < 0.001) (Figure 5C,D and Table 2). We also found a significant interaction between TGOT and dV/dt_min_ on Dur (i.e., a more pronounced increase in Dur by dV/dt_min_ in TGOT; *p* < 0.001) (Figure 5D and Table 2). Since there is evidence in the literature that AP initiation could be favored by a pronounced afterhyperpolarization with a regenerative mechanism [33], we constructed a LMM to test whether less-negative V_AHP_ values would influence V_thr_, but we found no evidence to support this hypothesis.

Could changes in AP parameters induced by TGOT determine the variations in the firing rate of the neurons observed during peptide administration? To answer this question, we constructed six LMMs to the study of the influence of (1) TGOT and Amp on ISI; (2) TGOT and V_AHP_ on ISI; (3) TGOT and Dur on ISI; (4) TGOT and dV/dt_max_ on ISI; (5) TGOT and dV/dt_min_ on ISI; (6) TGOT and V_thr_ on ISI. We found that none of these parameters influenced ISI, and there were no interactions with TGOT (Figure 6A–F, and Table 3) (*p* > 0.05). This result is of great interest because it implies the absence of causal dependencies between the AP shape and the firing frequency regarding the oxytocinergic modulation of OTR-expressing CA1 INs. Therefore, OTR activation may trigger parallel and independent mechanisms with different targets in these neurons, leading to the modification of their AP shape and to the increase in their firing rate, respectively. From this perspective, it can be hypothesized that TGOT-induced changes in the AP shape have specific and relevant functional roles in information processing in CA1 per se.

### 2.4. The Modifications of the AP Shape Induced by OTR Activation in the OTR-Expressing CA1 GABAergic Interneurons Do Not Result in the Increase in the Firing Frequency of These Neurons

The LMM results suggested that the changes in the AP shape and firing frequency induced by TGOT perfusion in the OTR-expressing CA1 INs are likely independent and not casually linked. These findings contrast those of a previous study concerning the oxytocinergic modulations of the CA2 pyramidal neurons, where the authors hypothesized that the increased neuronal firing rate would be a direct consequence of the TGOT-induced changes in AP shape [22].

To unequivocally demonstrate the direct modulatory effect of TGOT on the AP shape and its independence from the increased firing frequency, we set up a dedicated experiment on a subgroup of cells (*N* = 5 OTR-expressing INs from four mice). In this experiment, we used a feature of the Multiclamp 700B (see Section 4) to counteract the TGOT-induced membrane depolarization during the long recordings in current-clamp mode. In this way, we managed to maintain the membrane potential at a constant value, corresponding to the firing threshold of the neuron, while testing the effect of TGOT on the APs.

Under these conditions, we observed an evident effect on the peak of the APs during the bath perfusion of 1 µM TGOT, which was, however, reversed upon the wash out of the drug (decrease in V_peak_ in TGOT compared to the CTRL of 3.53 ± 1.58 mV vs. increase in V_peak_ in WASH compared to TGOT of 1.36 ± 0.61 mV; *N* = 5; *p* < 0.05) (Figure 7A). Importantly, the membrane potential remained relatively stable during the long recording (*p* > 0.05), confirming the efficacy of the recording protocol (Figure 7B), and the firing rate did not vary (*p* > 0.05) (Figure 7C). Significant changes in the AP shape during TGOT perfusion became evident through the phase–plane plot analysis (Figure 7D). Specifically, AP Amp was significantly reduced (47.8 ± 3.9 mV in the CTRL, reduced to 45.2 ± 3.6 mV in TGOT; *p* < 0.01), and V_AHP_ was modulated toward less-hyperpolarized values (−68.0 ± 0.8 mV in the CTRL vs. −66.9 ± 1.2 mV in TGOT; *p* < 0.05) (Figure 7E). On the other hand, V_thr_ remained unchanged (−48.6 ± 1.0 mV in CTRL vs. −48.6 ± 1.0 mV in TGOT; *p* > 0.05) (Figure 7E). We hypothesized that this parameter could be directly influenced by the TGOT-induced membrane depolarization and by the resultant increase in firing rate, due to the dynamic spike threshold phenomenon. This phenomenon involves a dynamic depolarization of V_thr_ during sustained firing at high frequency [34,35,36]. Consequently, it would be consistent for V_thr_ to not change during TGOT perfusion under this particular experimental protocol where the membrane potential and the firing rate were held constant. The dV/dt_max_ shifted from 91 ± 13 mV/ms to 83 ± 11 mV/ms (CTRL vs. TGOT; *p* < 0.05) and the dV/dt_min_ from −35.2 ± 6.6 mV/ms to −32.5 ± 6.2 mV/ms (CTRL vs. TGOT; *p* < 0.05) (Figure 7F), confirming a slowdown of AP kinetics induced by TGOT. Accordingly, AP Dur increased in the presence of TGOT (1.76 ± 0.31 ms in the CTRL vs. 1.83 ± 0.33 ms in TGOT; *p* < 0.05) (Figure 7G).

Overall, these findings establish that the modulation of the AP shape induced by TGOT occurs independent of membrane depolarization. Furthermore, they affirm that changes in the AP shape are not responsible for the TGOT-induced increase in the firing rate of the CA1 OTR-expressing INs, aligning with the predictions of the LMMs.

## 3. Discussion

In the hippocampus, OT plays crucial direct and indirect modulatory roles through highly diverse mechanisms specific to each subregion [14]. In the CA1, the activation of the OTR is known to increase the firing rate of specific classes of GABAergic INs [19,20,37]. This results in an indirect increase in the tonic and phasic inhibition on the pyramidal neurons [19] and in an enhanced ability to process relevant information by these neurons [20].

Our study provides novel evidence of the modulatory mechanisms through which OT acts on the CA1 GABAergic INs that express the OTR and are directly modulated by the peptide. Our data indeed demonstrate multiple modulatory effects exerted directly by the peptide on these neurons. Specifically, we found that OTR activation determines an increase in the firing rate of these neurons through two synergistic but independent mechanisms: (i) the depolarization of the membrane potential, which is intrinsically associated with an increase in the firing frequency, and (ii) the enhancement in cellular excitability, resulting in a higher firing rate in response to the same injected current. Similar results were obtained by Tirko et al. in the OTR-expressing pyramidal neurons of the CA2 region [22].

In addition, we found that OT can modify the shape of the APs by changing their threshold, reducing their amplitude and afterhyperpolarization, and slowing down their kinetics. These AP parameters are causally related to the number and the gating properties of the voltage-gated sodium and potassium channels that open during the AP, which are estimated to be significantly reduced during TGOT perfusion. Therefore, the opening of fewer channels that contribute to the rapid depolarization/repolarization phases of the AP likely results in slower kinetics and reduced AP amplitude and afterhyperpolarization. The presence of actual dependencies among the parameters describing the AP, which are varied by TGOT, was verified by specifically developed LMMs.

Although LMMs are not commonly utilized in basic neuroscience, their wider use in the analysis of complex data is strongly advocated by the scientific community [24]. This is because they guarantee rigor in the analysis, robustness, and reproducibility of the results, as well as the potential possibility of drawing broader and richer conclusions [24]. Through this approach, we were able to deduce that the modulatory action of OT is able to determine (i) a variation in the depolarization slope, consequently varying AP Amp; (ii) a variation in the repolarization slope, consequently varying V_AHP_; (iii) a change in the repolarization slope, thus varying AP Dur.

Regarding the TGOT-induced change in AP shape, there are again similarities with what was observed by Tirko et al. in the CA2 OTR-expressing pyramidal neurons; they found reductions in the AP overshoot and afterhyperpolarization following TGOT perfusion, but no changes in AP duration [22]. They concluded that a reduction in the peak of the depolarization phase of the AP should decrease the activation of the voltage-gated potassium channels that are typically opened during the repolarization phase of the AP, thus attenuating the post-spike afterhyperpolarization, favoring a rapid transition to the next spike, in turn [22]. Essentially, Tirko et al. hypothesized that TGOT-induced changes in AP shape in CA2 pyramidal neurons would result in an increased firing rate of these neurons. Our results indicate that this is not the case for the OTR-expressing GABAergic INs of the CA1 region. Using LMMs, we identified the absence of causal dependencies induced by TGOT perfusion between the variation in the parameters related to the AP shape and the increase in neuronal firing rate, assessed as the inverse of ISI. This evidence suggests that the two effects are driven by entirely independent mechanisms. The independence of the effects of TGOT on the AP shape and firing frequency was then experimentally confirmed, providing evidence of the predictive power of the LMMs.

Of particular interest is the presence of an OTR-mediated modulatory mechanism that specifically modifies the shape of the APs, without this being simply the first step toward alterations in the neuronal frequency coding. This suggests that the shape of AP plays a critical role in synaptic transmission to the principal neurons. The AP is usually considered as a purely digital event, but this idea should be abandoned [38]. In fact, the amplitude [39] and duration [40] of APs influence synaptic transmission. From this view, synaptic transmission is based on a hybrid between an “AP frequency code” (digital) and an “AP waveform code” (analog), called analog–digital synaptic transmission [38,41,42]. Our results indicate that OT is likely capable of modulating both these codes at the same time, finely modulating the synaptic transmission toward the principal neurons of CA1 in a highly reliable and informative manner.

The described oxytocinergic modulation of the spike shape and frequency of the CA1 OTR-expressing GABAergic interneurons would derive from the effect of the OTR activation on multiple and specific molecular targets—especially ion channels dynamics and the associated transmembrane currents—that however have not yet been clearly identified. A study conducted by Owen et al. suggested that the TGOT-induced effects on the OTR-expressing CA1 INs would involve the modulation of a mixed cationic current [20]. The results of Maniezzi et al. partially contradict this hypothesis, demonstrating instead the involvement of a calcium current, but not excluding the possible involvement of other ion currents as well [19]. In CA2, it was observed that OTR activation can modulate various currents of the OTR-expressing neurons at the same time [43]. The results of our study suggest that this is also the most plausible hypothesis for the OTR-expressing INs in the CA1 hippocampal region. Specifically, the activation of a calcium current might underlie the depolarization associated with the increase in firing rate, consistent with the previous findings of Maniezzi et al. [19]. At the same time, the negative modulation of an outward potassium current mediated by inward rectifier potassium channels (I_Kir_) could underlie the increase in neuronal excitability. Indeed, the literature data demonstrate that OTR activation can inhibit K_ir_ channels [44], and this same mechanism was recently reported in the CA2 region [43]. The negative modulation of voltage-gated sodium and potassium channels could explain the changes in the shape of the APs. In line with this hypothesis, OTR signaling was shown to modulate a voltage-gated sodium current in CA2 but with a positive effect [43], and Owen et al. demonstrated the involvement of sodium ions in TGOT-mediated effects on CA1 INs [20]. A possible modulation of the A-type potassium current (I_A_) might account for the slowdown in AP repolarization kinetics and the prolongation of the AP [45,46]. A modulatory effect of the OT on the I_A_ was demonstrated in the spinal cord [47]. However, further investigation and experimental validation are necessary to confirm these hypotheses.

In conclusion, OTR activation in OTR-expressing CA1 INs could trigger multiple signaling mechanisms proceedings along parallel pathways, thus leading to diverse modulatory effects not linked by cascading cause–effect relationships. However, these effects likely act synergistically toward a common outcome.

## 4. Materials and Methods

### 4.1. Animals and Brain Slices Preparation

All animal care and experimental procedures were conducted in compliance with EU directive 2010/63/EU and following relevant regulations and ethical standards defined by the Italian Legislative Decree No. 26 dated the 4th of March 2014.

Juvenile (P20-P30) heterozygous GAD67-GFP^+^ (Δneo) knock-in mice [48] were used. Before the experiment, mice were housed with food and water ad libitum, under a 12:12 h light/dark cycle. For the experiments, animals were anesthetized via inhalation of isoflurane and transcardially perfused with ice-cold (~4 °C), carboxygenated (95% O_2_–5% CO_2_) cutting solution, containing (in mM) sucrose 70; NaCl 80; KCl 2.5; NaHCO_3_ 26; glucose 15; MgCl_2_ 7; CaCl_2_ 1; NaH_2_PO_4_ 1.25. Following decapitation, the whole brain was removed, submerged into the ice-cold cutting solution, and sliced with a vibratome (DTK-1000, Dosaka EM, Kyoto, Japan). Transversal 300 µm thick slices containing the hippocampus were prepared [49]. Slices were then transferred to a recovery chamber filled with carboxygenated artificial cerebrospinal fluid (aCSF), containing (in mM) NaCl 125; KCl 2.5; NaHCO_3_ 26; glucose 15; MgCl_2_ 1.3; CaCl_2_ 2.3; NaH_2_PO_4_ 1.25. Slices were allowed to recover for 30 min at 37 °C and at least for 30 min at room temperature (~23 °C) before electrophysiological analyses.

### 4.2. Patch-Clamp Recordings

Electrophysiological recordings were performed at room temperature (~23 °C) on submerged slices perfused with carboxygenated aCSF at a rate of 0.8 to 1.4 mL/min. OTR was activated by the selective OTR agonist TGOT (Thr4,Gly7-oxytocin; Bachem, Bubendorf, Switzerland), which was dissolved in the aCSF at a final concentration of 1 µM and bath-perfused. The recording chamber was mounted on an E600FN microscope (Nikon, Tokyo, Japan) equipped with 4× and 40× water-immersion objectives and connected to a near-infrared CCD camera and fluorescence lamp/filters to allow visualization of cells. Whole-cell patch-clamp recordings were made from CA1 GAD67-GFP-expressing INs, located in the *stratum pyramidale*. The majority of experiments were performed in current-clamp mode. Patch pipettes were produced from borosilicate glass capillary tubes (Hilgenberg GmbH, Malsfeld, Germany) by using a horizontal puller (P-97, Sutter Instruments, Novato, CA, USA). They were filled with an intracellular solution iso-osmotic with cytosol, composed of (in mM) K-gluconate 130, NaCl 4, MgCl_2_ 2, EGTA 1, creatine phosphate 5, Na_2_ATP 2, Na_3_GTP 0.3, Hepes 10 (pH 7.3 with KOH). When filled with the above solution, patch pipettes had a resistance of 4–6 MΩ. Data were corrected offline for a liquid junction potential of +10.2 mV. Series resistance was minimized and monitored throughout the experiments. Signals were amplified with a MultiClamp 700B (Axon Instruments Molecular Devices, Sunnyvale, CA, USA), interfaced to a computer through a Digidata 1440 (Axon Instruments Molecular Devices, Sunnyvale, CA, USA), and acquired using Clampex 10.7 software (Molecular Devices, Palo Alto, CA, USA). Data were sampled at 20 kHz and filtered at 10 kHz.

### 4.3. Immunostaining of the Recorded Neurons

To stain the recorded cells, biocytin (3 mg/mL, Sigma-Aldrich, St. Louis, MO, USA) was added to the intracellular solution. Cells were held in whole-cell configuration for more than 15 min to allow biocytin diffusion into their cytosol. Following electrophysiological recordings, slices were (i) fixed in 4% paraformaldehyde for 1 h, (ii) rinsed with phosphate-buffered saline (PBS; Dulbecco’s, Sigma), (iii) rinsed alternately (16 washes of 10 min each) with quenching buffer (QB; glycine 0.1 M in PBS 120 mM) and blocking buffer (BB; BSA 1% and Triton X 0.3% in PBS 120 mM), (iv) incubated overnight with 5 µg/mL Alexa Fluor 568-conjugated streptavidin (Thermo Fisher Scientific, Waltham, MA, USA), (v) rinsed alternately (16 washes of 10 min each) with QB and BB, (vi) incubated for 30 min at 4 °C with 10 µg/mL DAPI (4′,6-diamidino-2-phenylindole; Molecular Probes), (vii) rinsed with PBS, (viii) mounted on microscope slides using ProLong TM glass antifade mountant (Thermo Fisher Scientific, Waltham, MA, USA), and (ix) stored in the dark at 4 °C until acquired via confocal microscopy (Leica SP8 STED 3x Confocal Microscope, Leica Microsystems, Wetzlar, Germany, and LAS X Life Science Software, version 3.7.4.23463).

### 4.4. Analysis of the Electrophysiological Recordings

Data were analyzed offline using Clampfit 10.7 (Molecular Devices, Palo Alto, CA, USA), Microcal OriginPro 2018 (OriginLab, Northampton, MA, USA), Microsoft Office Excel 365 (Microsoft, Redmond, WA, USA), and R-4.3.1 (R Core Team, Vienna, Austria).

The membrane capacitance (C_m_), the input resistance (R_in_), and the resting membrane potential (V_r_) were measured for each recorded cell. Following the establishment of a gigaseal, the membrane patch inside the pipette tip was broken to obtain the whole-cell configuration. This procedure was performed in voltage-clamp mode. For all the neurons, the first administered protocol was a voltage step from −70 mV to −80 mV in voltage-clamp mode. The peak of the capacitive current evoked by this −10 mV pulse was integrated to estimate the C_m_. The R_in_ was calculated following the same protocol as the ratio between the step of voltage (−10 mV) and the value of the current trace at the end of the 50 ms pulse, where the steady state was reached.

After switching to current-clamp mode, all subsequent experimental protocols were administered. Note that recordings in current-clamp mode obtained by using conventional patch-clamp amplifiers could be affected by predictable and unpredictable errors, especially impacting rapid events such as APs [50]. These technical issues are mainly due to the conventional electronic design of the patch-clamp headstages and can be overcome by using an headstage that integrates both a current-to-voltage converter for voltage-clamp and a voltage follower for current-clamp [50], such as the CV-7B headstage of the Multiclamp 700B amplifier used in this study.

Initially, V_r_ was detected with 0 pA current injection. Then, the effect of TGOT on the membrane potential and firing frequency was assessed through a current-clamp long-recording protocol, with the imposition of a current value capable to evoke a stable, just-suprathreshold, repetitive firing for each cell. R_in_ was monitored throughout recordings via brief hyperpolarizing current pulses (in current-clamp mode, without switching back to voltage-clamp mode) and was found to remain stable over time. The quantification of the TGOT-induced effect was performed as previously described [19]. Briefly, all-point histograms for 10 s long intervals recorded under control (CTRL) conditions and during TGOT perfusion were fitted with gaussian functions. The difference between the membrane potentials at the gaussian peak in TGOT and CTRL provided the value of depolarization (∆V) for each cell. The ratio between the number of the APs in the same 10 s intervals in TGOT and CTRL provided the increase in the normalized firing frequency for each cell.

In current-clamp mode, the firing rate of the OTR-expressing neurons in response to depolarizing currents steps of increasing intensity was recorded, before and after TGOT administration. Firing rate-to-injected current (F–I) relationships [19,51,52,53] were generated and analyzed to evaluate possible changes in the offset (i.e., the minimal intensity of injected current required to attain a response) and the gain (i.e., the slope of the relationship), accounting for effects of TGOT on cell excitability.

To designate the features of the APs in CTRL and TGOT, multiple quantitative parameters characterizing the AP waveform were extracted and compared via phase–plot analysis [27,28]. Starting from the previously described long-recording protocol, 10 consecutive APs in CTRL and 10 consecutive APs in TGOT were selected for each cell. On a subgroup of cells, a modified long-recording protocol was used, designed to counteract the TGOT-induced depolarization: to this end, we used a Multiclamp 700B feature that injects a slow current into the cell to maintain the membrane potential at a constant value. Changes in the first derivative of the membrane potential with respect to time (dV/dt expressed in mV/ms) were plotted against the instantaneous value of the membrane potential itself (expressed in mV). The resulting phase–plane plot was used to extract the AP threshold (V_thr_—calculated using the maximum second derivative in the phase–plane plot method as described in Sekerli et al., 2004 [54]), the AP amplitude (Amp—difference between the extreme right voltage value assumed by the phase–plane plot and the V_thr_), the peak voltage value of the AP afterhyperpolarization (V_AHP_—extreme left voltage value assumed by the phase–plane plot), the AP depolarization slope/kinetics (dV/dt_max_—maximal dV/dt value assumed by the upper part of the phase–plane plot), and the AP repolarization slope/kinetics (dV/dt_min_—minimal dV/dt value assumed by the lower part of the phase–plane plot). The AP duration, not obtainable from phase plot analysis, was calculated directly on the AP waveform as the spike width (in ms) measured at half-maximal spike amplitude. The net ionic current (I_ionic_) through the membrane during the AP was estimated using the following equation: I_ionic_ = −C_m_ × (dV/dt) [28,29], where C_m_ is the membrane capacitance of the cell estimated in voltage-clamp mode, as described at the beginning of this paragraph, and the (dV/dt) is the variation in the voltage over time during the APs recorded in current-clamp mode.

To correlate the parameters describing the AP waveform with the firing frequency, a one-to-one correspondence between the parameters to be compared would be needed. Therefore, for this purpose, we evaluated the firing frequency by computing the interspike intervals (ISIs) between the contiguous APs used in the phase plot analysis, which are the inverse of the instantaneous firing frequency.

### 4.5. Statistics and Linear Mixed-Effects Models

Statistical analyses and plots were performed and drawn with Microsoft Office Excel 365, Microcal OriginPro 2018, and R-4.3.1. Data throughout the text are expressed as all-point plots together with summary statistics (mean ± standard error of the mean (SEM)). N indicates the number of statistical units analyzed for each experimental procedure, as detailed in the results. Statistical significance was determined via a paired two-tailed Student’s *t*-test or one sample *t*-test, according to the type of experiment. To assess linear pairwise relationships in the parameters under investigation, we analyzed their scatterplot matrices and computed the correspondent Pearson correlation coefficients and significance values, as in Binini et al., 2021 [55].

The linear mixed-effects models (LMMs) were constructed in the R environment using the *nlme* [56] and *psych* [57] packages. The interclass correlation coefficient (ICC) was used to determine the correlation within the cluster. Since (i) each cell could be considered as an experimental unit randomly drawn from a general population, and (ii) TGOT could have an effect of different intensity on each cell, models were constructed using (i) a random intercept related to each considered cell (i.e., the intercepts are allowed to vary so that the predicted scores on the dependent variable also varied for each cell) and (ii) a random slope for each experimental condition (i.e., the two groups—CTRL and TGOT—can have a different slope, allowing the independent variable to have a different effect for each group). The influence of these random effects was evaluated by comparing the constructed models and general linear models. As indicated in the literature [26], models were chosen as they had the lowest Akaike information criterion (AIC) and Bayesian information criterion (BIC), computed using the function *anova.lme* in the *nlme* package.

To evaluate the effect of TGOT on each analyzed AP parameter and on the ISI, models were constructed using the considered AP parameter (or ISI) as the dependent variable and the experimental condition as the fixed effect. The influence of the fixed effect was also assessed using the *anova* function in the *nlme* package, which returned a *p*-value for the Wald test [58]. Other models were constructed to examine the interaction between TGOT and a selected AP parameter relative to each other (experimental condition and the parameter were entered in the model as fixed effects) and their influence on an additional AP parameter (or ISI) (dependent variable in the model). For higher-accuracy models, the AP parameters used as fixed effects were first centered, subtracting the mean value obtained in the control condition. The results obtained with all the models were fitted with the restricted maximum likelihood (REML) method [26], given the number of the samples and the complexity of the random effects. The goodness of each model was evaluated by checking the normality of the random effects and the residual distribution using the *qqnorm* function in the *stats* package. The variance of the random effects and the marginal and conditional R^2^ reported in Table 1, Table 2 and Table 3 were calculated using the *sjPlot* package [59]. Graphs used for visualizing the reported LMMs were constructed using the *ggplot2* package [60].

## Figures and Tables

**Figure 1 ijms-25-02613-f001:**
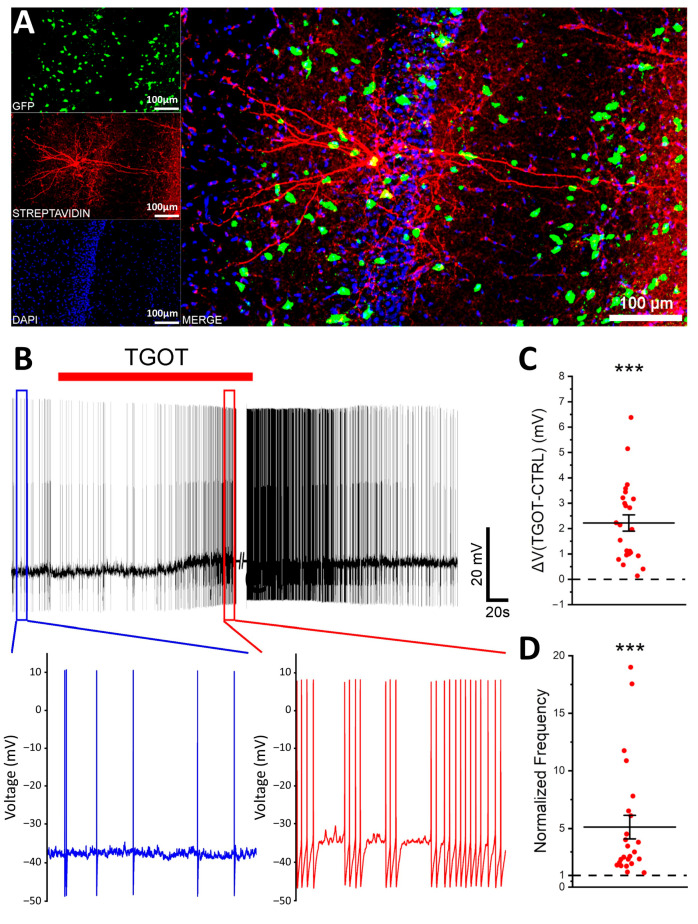
TGOT depolarizes the membrane potential and increases the firing rate of the OTR-expressing CA1 GABAergic INs. (**A**) Confocal images showing the GFP^+^ cells (green); the recorded IN marked with Alexa Fluor 568-conjugated streptavidin labeling of biocytin (red); all the cell nuclei marked with DAPI (blue); the colocalization of GFP (green), Alexa Fluor 568-conjugated streptavidin used to label biocytin (red), and DAPI (blue) that allowed confirmation of the location in the *stratum pyramidale* of the CA1 hippocampal region and the GABAergic identity of the recorded cell. (**B**) Representative voltage trace at spike threshold, showing the response of an OTR-expressing CA1 GABAergic IN to the administration of 1 µM TGOT (red bar). The insets show the magnification of representative 10 s long control (blue) and TGOT (red) tracts of the trace. (**C**,**D**) All-point plots together with summary statistics (mean ± SEM) showing the depolarization (**C**) and the increase in the spike frequency (**D**) induced by TGOT (*N* = 24 cells from 18 mice; one-sample *t*-test, *** *p* < 0.001).

**Figure 2 ijms-25-02613-f002:**
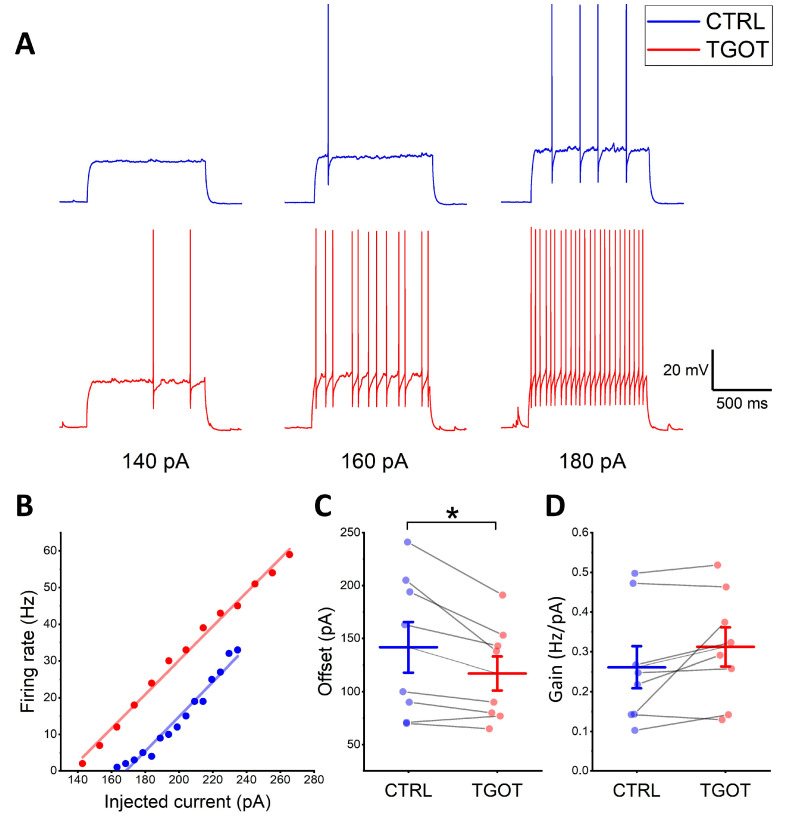
TGOT enhances the excitability of the OTR-expressing CA1 GABAergic INs. (**A**) Representative voltage traces recorded from an OTR-expressing CA1 GABAergic IN in response to the injection of depolarizing current steps of increasing amplitude (140, 160, and 180 pA) starting from –70 mV. Traces recorded in control (CTRL) are in blue, and traces recorded in TGOT (TGOT) are in red. Notice the higher number of action potentials in TGOT compared to CTRL at all current injections. (**B**) Firing-rate-to-injected current (F–I) relationships referred to the traces of (**A**) in CTRL (blue) and TGOT (red), fitted with linear regression functions. (**C**,**D**) All-point plots together with summary statistics (mean ± SEM) comparing the values of the offset (**C**) and the gain (**D**) obtained from the F–I relationships in CTRL and TGOT (*N* = 8 cells from 5 mice; paired *t*-test, * *p* < 0.05).

**Figure 3 ijms-25-02613-f003:**
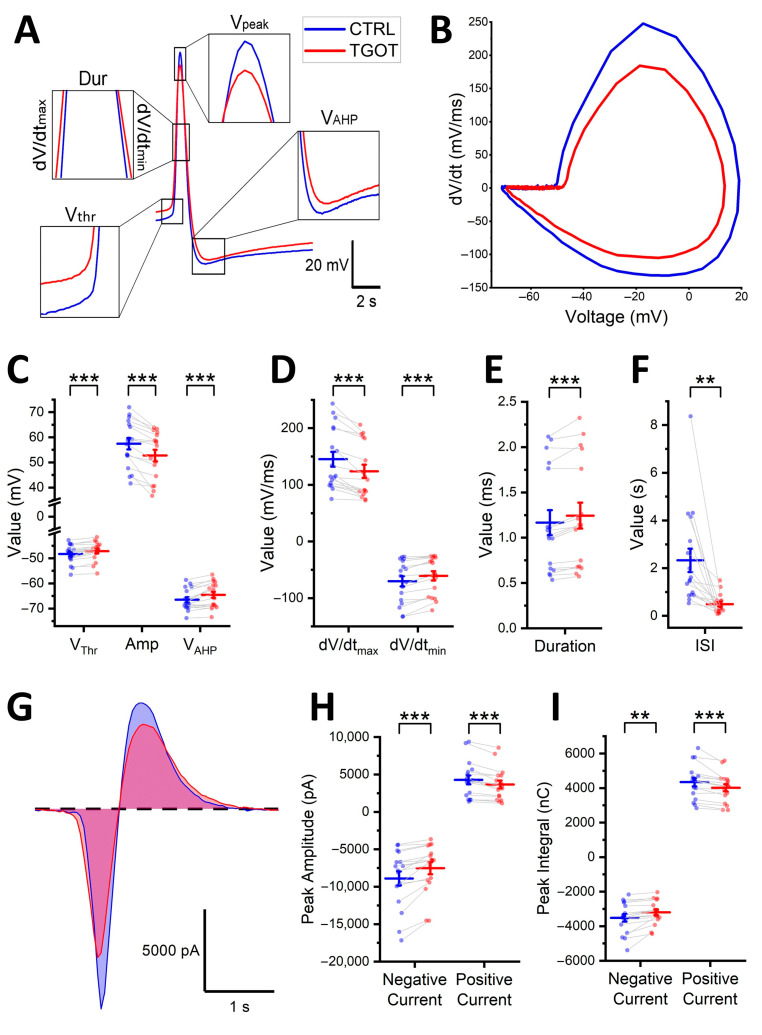
TGOT modifies the shape of the APs of the CA1 INs directly modulated by it. (**A**) Representative AP waveforms in CTRL (blue) and TGOT (red) overlapped to show the variations (magnified in the insets) in the threshold, amplitude, afterhyperpolarization, and kinetics induced by TGOT. (**B**) Phase–plane plots of the APs shown in (**A**). (**C**,**E**) All-point plots together with summary statistics (mean ± SEM) comparing the values of V_thr_, Amp, V_AHP_ (**C**), dV/dt_max_ and dV/dt_min_ (**D**), and Dur (**E**) in CTRL and TGOT (*N* = 17 cells from 12 mice; paired *t*-test, ** *p* < 0.01; *** *p* < 0.001). (**F**) All-point plot together with summary statistics (mean ± SEM) comparing the values of ISI in CTRL and TGOT (*N* = 17 cells from 12 mice; paired *t*-test, ** *p* < 0.01; *** *p* < 0.001). (**G**) Net ionic current estimated from the AP waveforms in (**A**) in CTRL (blue) and TGOT (red). Notice the reduction in both the inward current and the outward current in TGOT. (**H**,**I**) All-point plots together with summary statistics (mean ± SEM) comparing the amplitude (**H**) and the integral (**I**) of the inward and the outward currents during the APs in CTRL and TGOT (*N* = 17 cells from 12 mice; paired *t*-test, ** *p* < 0.01; *** *p* < 0.001).

**Figure 4 ijms-25-02613-f004:**
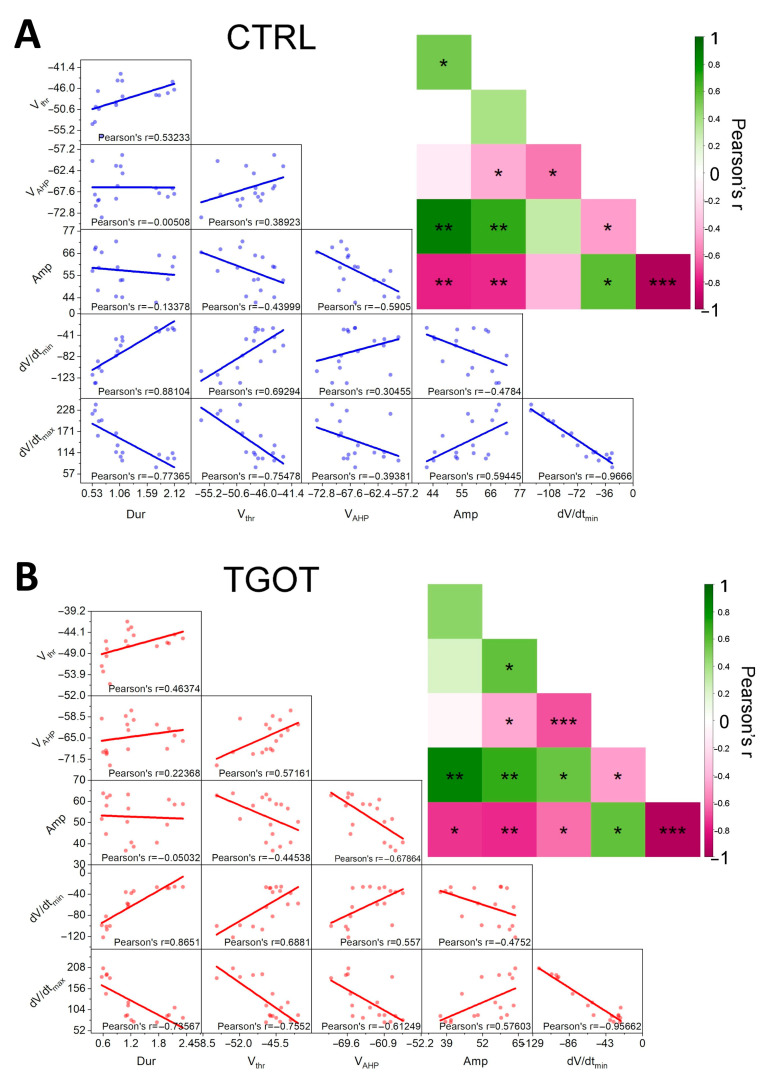
Many AP parameters correlated between each other in the analyzed CA1 GABAergic INs, and these correlations strengthened in the presence of TGOT. (**A**) In control conditions, pairwise scatterplot matrices, interpolated with regression lines, together with Pearson correlation coefficients for V_thr_ (mV), V_AHP_ (mV), Amp (mV), dV/dt_min_ (mV/ms), dV/dt_max_ (mV/ms), and Dur (ms) (N = 17 cells from 12 mice). The same matrices overlaid on the corresponding color-coded correlation matrices together with the significance value associated with each scatterplot (* *p* < 0.05; ** *p* < 0.01; *** *p* < 0.001) are also reported. (**B**) Same as (**A**) for data obtained during TGOT perfusion.

**Figure 5 ijms-25-02613-f005:**
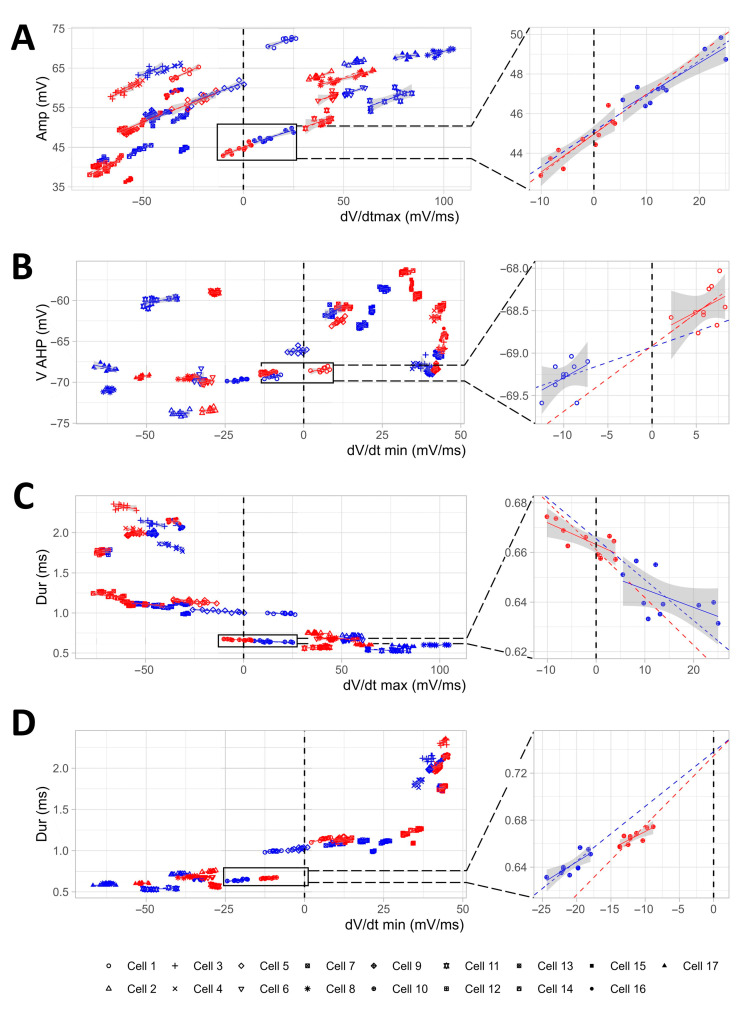
LMMs indicate that TGOT interacts with most of the considered AP parameters to strengthen their influence on each other. (**A**–**D**) The graphs help to visualize the constructed LMMs used to study the effects and interaction of TGOT and dV/dt_max_ on Amp (**A**), TGOT and dV/dt_min_ on V_AHP_ (**B**), TGOT and dV/dt_max_ and Dur (**C**), and TGOT and dV/dt_min_ on Dur (**D**). Each symbol is related to a specific cell (*N* = 17 cells from 12 mice), and the blue and red colors refer to the CTRL and TGOT conditions, respectively. The dashed line represents the interpolation predicted by the LMM, while the continuous line is the real interpolation for each cell in each experimental condition. The gray area indicates the 95% confidence interval. The vertical dashed line at x = 0 (which corresponds to the mean of the independent variable computed in CTRL condition) illustrates the ordinate axis used by the model. On the right of each graph, an enlargement of a representative cell is shown. The change in the slope of the dashed line in TGOT with respect to CTRL represents the effect of the interaction between TGOT and the considered AP parameters on the dependent variable.

**Figure 6 ijms-25-02613-f006:**
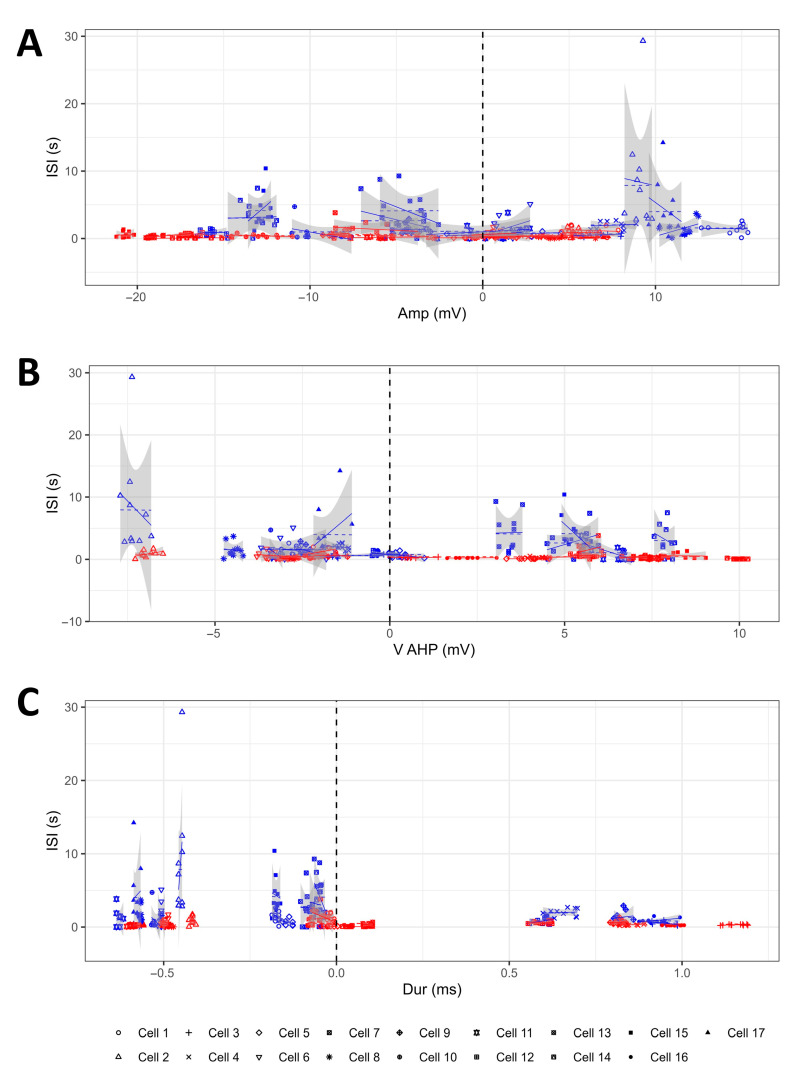
No significant interaction between TGOT and AP parameters existed to determine the ISI reduction during TGOT perfusion. (**A**–**F**) The graphs help to visualize the constructed LMMs used to study the effects and interaction of TGOT and Amp on ISI (**A**), TGOT and V_AHP_ on ISI (**B**), TGOT and Dur on ISI (**C**), TGOT and dV/dt_max_ on ISI (**D**), TGOT and dV/dt_min_ on ISI (**E**), and TGOT and V_thr_ on ISI (**F**). Each symbol is related to a specific cell (*N* = 17 cells from 12 mice), and the blue and red colors refer to the CTRL and TGOT conditions, respectively. The dashed line represents the interpolation predicted by the LMM, while the continuous line is the real interpolation for each cell in each experimental condition. The gray area indicates the 95% confidence interval. The vertical dashed line at x = 0 (which correspond to the mean of the independent variable computed in CTRL condition) illustrates the ordinate axis used by the model.

**Figure 7 ijms-25-02613-f007:**
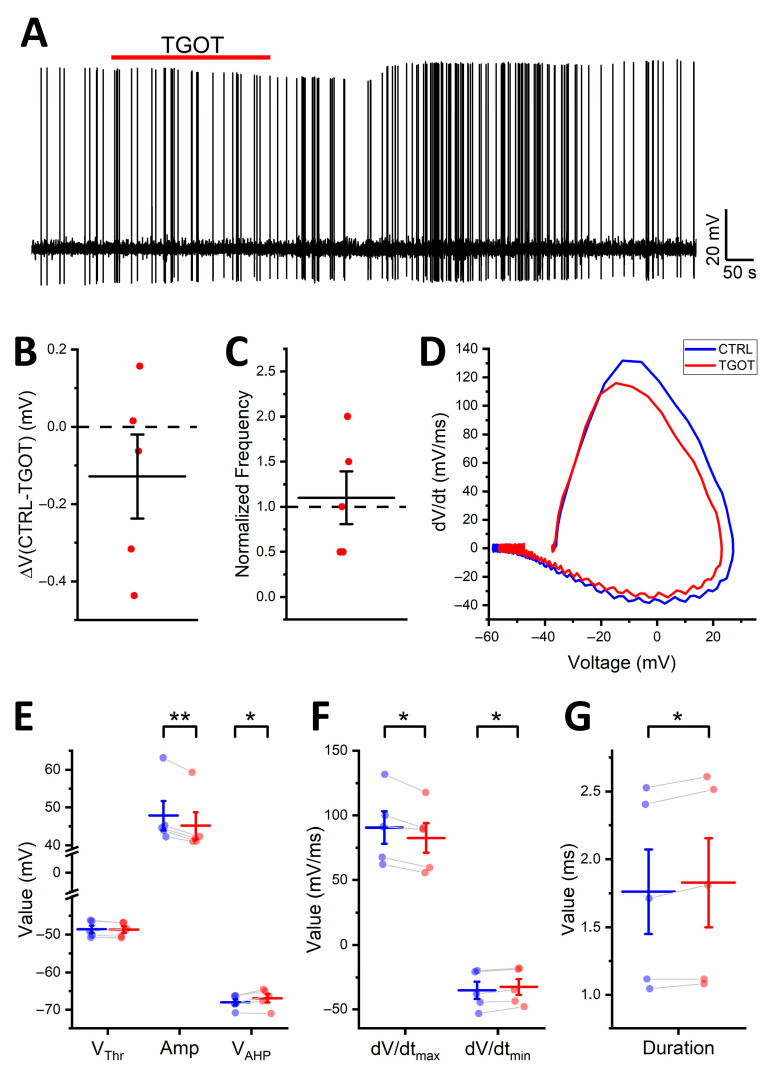
In the absence of membrane depolarization, TGOT modifies the shape of the APs but not the firing rate of the CA1 INs directly modulated by it. (**A**) Representative voltage trace at a constant spike threshold level showing the response of an OTR-expressing CA1 GABAergic IN to the administration of 1 µM TGOT (red bar). (**B**,**C**) All-point plots together with summary statistics (mean ± SEM) showing the absence of depolarization (**B**) and no increase in the spike frequency (**C**) induced by TGOT (*N* = 5 cells from 4 mice; one-sample *t*-test, *p* > 0.05). (**D**) Phase–plane plots of two representative APs in CTRL (blue) and TGOT (red). (**E**–**G**) All-point plots together with summary statistics (mean ± SEM) comparing the values of the V_thr_, Amp, V_AHP_ (**E**), dV/dt_max_ and dV/dt_min_ (**F**), and Dur (**G**) in CTRL and TGOT (*N* = 5 cells from 4 mice; paired *t*-test, * *p* < 0.05; ** *p* < 0.01).

**Table 1 ijms-25-02613-t001:** LMMs confirmed a significant effect of TGOT on V_thr_, Amp, V_AHP_, dV/dt_max_, dV/dt_min_, Dur, and ISI.

	CTRL	TGOT	ICC	*p*-Value from *t*-Test	*p*-Value from LMMs
V_thr_	−48.3 ± 0.9 mV	−47.1 ± 0.9 mV	0.96	<0.001	<0.001
Amp	57.4 ± 2.3 mV	52.7 ± 2.3 mV	0.99	<0.001	<0.001
V_AHP_	−66.5 ± 1.0 mV	−64.6 ± 1.2 mV	1.00	<0.001	<0.001
dV/dt_max_	145 ± 13 mV/ms	124 ± 12 mV/ms	0.99	<0.001	<0.001
dV/dt_min_	−70.2 ± 9.2 mV/ms	−60.8 ± 8.2 mV/ms	1.00	<0.001	<0.001
Dur	1.17 ± 0.14 ms	1.24 ± 0.14 ms	1.00	<0.001	<0.001
ISI	2.33 ± 0.49 s	0.49 ± 0.10 s	0.35	<0.01	<0.001

**Table 2 ijms-25-02613-t002:** LMMs results for effects of TGOT and AP parameters on each other.

	Effect of TGOT and dV/dt_max_ on Amp (mV)			Effect of TGOT and dV/dt_min_ on V_AHP_ (mV)			Effect of TGOT and dV/dt_max_ on Dur (ms)			Effect of TGOT and dV/dt_min_ on Dur (ms)		
Predictors	Estimates	Std. Error	*p*	Estimates	Std. Error	*p*	Estimates	Std. Error	*p*	Estimates	Std. Error	*p*
(Intercept)	57.4349 ***	2.0727	**<0.001**	−66.4859 ***	0.9782	**<0.001**	1.1668 ***	0.1211	**<0.001**	1.1668 ***	0.1016	<0.001
Effect of dV/dt_max_ in CTRL	0.1781 ***	0.0077	**<0.001**				−0.0016 ***	0.0002	**<0.001**			
Effect of TGOT at x = 0	−0.3996	0.4471	0.372	1.2087 **	0.3655	**0.001**	0.0363 *	0.0151	**0.017**	0.0216	0.0144	0.134
Interaction of dV/dt_max_ and TGOT	0.0243 ***	0.0073	**0.001**				−0.0003	0.0002	0.209			
Effect of dV/dt_min_ in CTRL				0.0364 **	0.0111	**0.001**				0.0047 ***	0.0006	**<0.001**
Interaction of dV/dt_min_ and TGOT				0.0407 ***	0.0094	**<0.001**				0.0013 ***	0.0004	**0.001**
**Random Effects**												
σ^2^	0.32			0.09			0.00			0.00		
τ_00_	73.00 _cell_			16.26 _cell_			0.25 _cell_			0.18 _cell_		
τ_11_	2.75 _cell_._cond(TGOT)_			1.99 _cell_._cond(TGOT)_			0.00 _cell_._cond(TGOT)_			0.00 _cell_._cond(TGOT)_		
ρ_01_	0.10 _cell_			−0.06 cell			0.37 _cell_			0.09 _cell_		
ICC	1.00			0.99			1.00			1.00		
N	17 _cell_			17 _cell_			17 _cell_			17 _cell_		
Observations	340			340			340			340		
Marginal R^2^/Conditional R^2^	0.554/0.998			0.230/0.996			0.034/0.999			0.164/0.999		

For each fixed effect, the predicted value, the standard error, and the associated *p*-value are reported (* *p* < 0.05; ** *p* < 0.01; *** *p* < 0.001; significant values are reported in bold). The intercept indicates the mean value of the dependent variable in CTRL condition. The effect of dV/dt_max_ and dV/dt_min_ in CTRL condition shows how much the dependent variable increases or decreases for each unit of the independent variable in control condition. The effect of TGOT at x = 0 indicates the difference in the dependent variable in CTRL and TGOT condition when x is equal to 0 (i.e., when the independent variable is equal to the mean calculated in control condition). The interaction of TGOT and the independent variable shows if the interaction is constructive or destructive (shows how much the slope varies between CTRL and TGOT conditions). At the bottom of the table, the random-effects variances are reported. σ^2^ is the residual variance, which indicates the within-cell variance. τ_00_ and τ_01_ are the random intercept variance (that indicates how much the cells differ to each other) and the random slope variance, respectively. ρ_01_ is the random slope–intercept correlation (that is, the correlation between the random slope and the random intercept). The marginal R^2^ (i.e., the R^2^ calculated considering only the fixed-effects variance), and the conditional R^2^ (which instead takes both the fixed and random effects into account) are reported too.

**Table 3 ijms-25-02613-t003:** LMMs results for effects of TGOT and AP parameters on ISI.

	Effect of TGOT and Amp on ISI (s)		Effect of TGOT and V_AHP_ on ISI (s)		Effect of TGOT and Dur on ISI (s)		Effect of TGOT and dV/dt_max_ on ISI (s)		Effect of TGOT and dV/dt_min_ on ISI (s)		Effect of TGOT and Dur on ISI (s)	
Predictors	Est. ± SE	*p*	Est. ± SE	*p*	Est. ± SE	*p*	Est. ± SE	*p*	Est. ± SE	*p*	Est. ± SE	*p*
(Intercept)	2.33 ± 0.50 ***	**<0.001**	2.33 ± 0.50 ***	**<0.001**	2.33 ± 0.50 ***	**<0.001**	2.33 ± 0.50 ***	**<0.001**	2.33 ± 0.50 ***	**<0.001**	2.33 ± 0.50 ***	**<0.001**
Effect of Amp in CTRL	0.01 ± 0.05	0.881										
Effect of TGOT at x = 0	−1.83 ± 0.51 ***	**<0.001**	−1.81 ± 0.51 ***	**<0.001**	−1.83 ± 0.48 ***	**<0.001**	−1.86 ± 0.50 ***	**<0.001**	−1.84 ± 0.49 ***	**<0.001**	−1.84 ± 0.44 ***	**<0.001**
Interaction Amp and TGOT	−0.01 ± 0.05	0.919										
Effect of V_AHP_ in CTRL			−0.06 ± 0.12	0.606								
Interaction of V_AHP_ and TGOT			0.05 ± 0.12	0.700								
Effect of Dur in CTRL					−1.24 ± 0.86	0.150						
Interaction of Dur and TGOT					1.21 ± 0.87	0.165						
Effect of dV/dt_max_ in CTRL							0.01 ± 0.01	0.480				
Interaction of dV/dt_max_ and TGOT							−0.01 ± 0.01	0.407				
Effect of dV/dt_min_ in CTRL									−0.01 ± 0.01	0.301		
Interaction of dV/dt_min_ and TGOT									0.01 ± 0.01	0.303		
Effect of V_thr_ in CTRL											−0.19 ± 0.10	0.073
Interaction of V_thr_ and TGOT											0.19 ± 0.11	0.070
**Random Effects**												
σ^2^	3.52		3.51		3.51		3.51		3.51		3.54	
τ_00_	3.95 _cell_		3.95 _cell_		3.47 _cell_		3.80 _cell_		3.70 _cell_		2.88 _cell_	
τ_11_	3.64 _cell.cond(TGOT)_		3.68 _cell.cond(TGOT)_		3.18 _cell.cond(TGOT)_		3.45 _cell.cond(TGOT)_		3.37 _cell.cond(TGOT)_		2.55 _cell.cond(TGOT)_	
ρ_01_	−1.00 _cell_		−1.00 _cell_		−1.00 _cell_		−1.00 _cell_		−1.00 _cell_		−1.00 _cell_	
ICC	0.36		0.36		0.33		0.35		0.35		0.29	
N	17 _cell_		17 _cell_		17 _cell_		17 _cell_		17 _cell_		17 _cell_	
Observations	340		340		340		340		340		340	
Marginal R^2^/Conditional R^2^	0.134/0.447		0.138/0.450		0.170/0.446		0.143/0.445		0.153/0.447		0.179/0.417	

At the top of the table, the predicted value, the standard error, and the associated *p*-value for each fixed effect are reported (*** *p* < 0.001; significant values are reported in bold). At the bottom of the table, the random-effects variances and the marginal and conditional R^2^ are shown. For the details and the interpretation of the single parameters, refer to Table 2.

## Data Availability

The raw data supporting the conclusions of this article will be made available by the authors upon request.

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
