# Peer review of "Oxytocin Modifies the Excitability and the Action Potential Shape of the Hippocampal CA1 GABAergic Interneurons"

_ijms, 2024, doi:10.3390/ijms25052613_

Round 1

Reviewer 1 Report

Comments and Suggestions for Authors

This manuscript by Catagno et al uses whole cell patch clamp recordings to carefully evaluate the effect of TGOT on GAD67 positive interneurons in the CA1 pyramidal cell layer. Prior studies have indicated these cells are excited by TGOT, but as noted by the authors, then focused largely on downstream effects on the pyramidal cells themselves. The current manuscript aspires to be the first to carefully evaluate OTR coupled effectors modulated by bath applied TGOT specifically in these interneurons.

The data suggest that TGOT (as expected) depolarizes the cells and increases the spontaneous firing rate. The authors also look carefully at action potential kinetics where they conclude TGOT also decreases maximum rate of depolarization and repolarization, decreases peak amplitude, and decreases the AHP. Finally, they perform an extensive analysis using LMM techniques. This analysis reveals the extent to which various aspects of the action potential kinetics correlate with one another both before and after TGOT application. A number of the correlations observed in control conditions are self-evident (e.g. slower dV/dt_min or dV/dt_max correlate strongly with increased AP duration). However, the fact that some correlations but not others are strengthened after application of TGOT (particularly those that involve the AHP) is used to argue that TGOT mediated effects on fast action potential kinetics occur via a mechanism that is independent of effects on membrane potential and spontaneous firing rate.

Overall, the manuscript is interesting, the data appear to be of high quality, and the analytic work is carefully done. I support and appreciate detailed analysis and consideration of OTR mediated effects on cell physiology. It seems plausible that the core conclusion that multiple effectors are targeted independently is correct.

That said I have several significant concerns about the study that limit enthusiasm substantially.

Major Comments:

Given the that the sodium and potassium channels underlying fast action potential kinetics and the AHP are so highly voltage sensitive, and that TGOT clearly depolarizes INs, it seems problematic that no effort was made to measure fast AP kinetics (and the AHP) both before and after TGOT at an identical transmembrane voltage.

Given that the core effects on kinetics and the AHP (slower, wider, shorter/smaller) are all consistent with run down, and that it takes time to bath apply TGOT, the study really needs some time matched controls, and ideally an experiment that involves an OTR antagonist as well.

Minor comments

Methods state the input resistance was continuously monitored, but effects of TGOT on Rin are not clearly reported.

Major findings are reported with N indicating the number of cells reported. It would also be helpful to report the number of animals represented in each major experiment.

Line 263: Please clarify why only a subset of cells (8 of 19) were analyzed for Figure 2.

118: “to allow cells visualization” should be “to allow visualization of cells”

134: This paragraph is bold

172: Here “offset” is used to describe the current required to elicit an AP. Consider using the term “rheobase” throughout the paper.

348: This sentence likely belongs to the previous paragraph. Similar suggestion for line 364.

351: The word “Then” may be omitted.

416: Clarify that this is an R package described in Methods

Figure 1A: It would be helpful if something could be done to improve the quality of this figure. Red-on-black compresses poorly in PDFs and is not color-blind friendly.

Figure 1: D reports AP frequency (Hz) but most subsequent figures report ISI (ms). Consider adding a ΔISI (ms) plot to this figure.

Figure 3: Consider hiding bar outlines to be consistent with Figure 2

Figure 4: Would benefit from titles next to A and B indicating “CTRL” and “TGOT” similar to Figures 2-3

Figure 4: It would be helpful if color bars had an axis label like “Correlation” or “Pearson’s r”.

Figure 5: “strength” should be “strengthen” in line 471

Figures 4-6: N should be reported in the legend.

Reviewer 2 Report

Comments and Suggestions for Authors

Please see attached

Comments on the Quality of English Language

non-available

Round 2

Reviewer 1 Report

Comments and Suggestions for Authors

The authors made revisions that address most of my comments. Please see remaining comments below: 

Line 203: To say time matched controls did not show any noteworthy differences seems not correct given that in their response to reviews the authors indicate that Vpeak was decreased by 1.54 +/- 0.53 mV with a p-value < 0.05 over a 200 sec period when TGOT was not applied. In the response to reviews the authors make the argument that although this change is significant, it’s not as big as the TGOT induced change. They should do the same in the manuscript (rather than suggest time match controls showed ‘no noteworthy differences’).   

Similarly, the authors suggest in the response to reviews that recovery of the effects of TGOT on washout argues against rundown. This could be a powerful argument, but they’d have to actually analyze and quantify recovery during washout across multiple cells rather than highlighting a single raw data trace absent quantitative analysis in any/all cells. If the data show a consistent / quantifiable recovery, I think this would definitely be worth adding to the manuscript. If not, then subjective comments about representative traces should be removed. 

Line 401:  “this parameter (Vthrehold) can be directly influenced by depolarization, so this result was expected given the peculiar experimental protocol.”  There may not be a super clear revision to make for this comment, but for whatever it is worth, I think it’s actually more surprising that there is an effect on Vthreshold in Fig. 3C than that there is not in Fig. 7.  Threshold depends heavily on voltage sensitivity, and minimally on overall conductance, of voltage gated sodium channels, so when line 401 says threshold can be directly influenced by depolarization that sounds unlikely / wrong to me. I suspect the apparent effect on Vthreshold in Fig. 3 is a kind of misdirect that comes largely from the fact that all the analysis is being done on spontaneous action potentials occurring at slightly different membrane potentials. A much better way to measure threshold is to evaluate it on the first action potential observed during a slow current ramp that goes from something negative of rest to something positive of threshold. I’d suspect TGOT has no effect on that.

Reviewer 2 Report

Comments and Suggestions for Authors

Comments on the Quality of English Language

non-available

Round 3

Reviewer 2 Report

Comments and Suggestions for Authors

Comments on the Quality of English Language

non-available
